# Identification of Genes Associated with Resistance to Persulcatusin, a Tick Defensin from *Ixodes persulcatus*

**DOI:** 10.3390/microorganisms12020412

**Published:** 2024-02-19

**Authors:** So Shimoda, Junya Ito, Tasuke Ando, Ryuta Tobe, Kiyotaka Nakagawa, Hiroshi Yoneyama

**Affiliations:** 1Laboratory of Animal Microbiology, Department of Animal Science, Graduate School of Agricultural Science, Tohoku University, 468-1, Aramaki Aza Aoba, Aoba-ku, Sendai 980-0845, Japan; so.shimoda.t7@dc.tohoku.ac.jp (S.S.); tasuke.ando.d4@tohoku.ac.jp (T.A.); ryuta.tobe.c7@tohoku.ac.jp (R.T.); 2Laboratory of Food and Biodynamic Chemistry, Graduate School of Agricultural Science, Tohoku University, 468-1, Aramaki Aza Aoba, Aoba-ku, Sendai 980-0845, Japan; junya.ito.d3@tohoku.ac.jp (J.I.); kiyotaka.nakagawa.c1@tohoku.ac.jp (K.N.)

**Keywords:** antimicrobial peptides, *Staphylococcus aureus*, persulcatusin

## Abstract

Antimicrobial peptides (AMPs) are present in a wide range of plants, animals, and microorganisms. Since AMPs are characterized by their effectiveness against emergent antibiotic-resistant bacteria, they are attracting attention as next-generation antimicrobial compounds that could solve the problem of drug-resistant bacteria. Persulcatusin (IP), an antibacterial peptide derived from the hard tick *Ixodes persulcatus*, shows high antibacterial activity against various Gram- positive bacteria as well as multidrug-resistant bacteria. However, reports on the antibacterial action and resistance mechanisms of IP are scarce. In this study, we spontaneously generated mutants showing increased a minimum inhibitory concentration (MIC) of IP and analyzed their cross-resistance to other AMPs and antibiotics. We also used fluorescent probes to investigate the target of IP activity by evaluating IP-induced damage to the bacterial cytoplasmic membrane. Our findings suggest that the antimicrobial activity of IP on bacterial cytoplasmic membranes occurs via a mechanism of action different from that of known AMPs. Furthermore, we screened for mutants with high susceptibility to IP using a transposon mutant library and identified 16 genes involved in IP resistance. Our results indicate that IP, like other AMPs, depolarizes the bacterial cytoplasmic membrane, but it may also alter membrane structure and inhibit cell-wall synthesis.

## 1. Introduction

Antimicrobial peptides (AMPs) are typically short cationic amphiphilic peptides with broad-spectrum antimicrobial activity and are an important component of innate immunity in all organisms [1]. Their antimicrobial activity is due to the disruption of the bacterial cytoplasmic membrane following the interaction between the cationic amphiphilic AMPs and the negatively charged bacterial cytoplasmic membrane [2]. AMPs exert antimicrobial activity through multiple mechanisms via interaction with their targets. For example, the interaction of some AMPs with cell-wall precursors inhibits cell-wall biosynthesis [3,4] and interaction of the AMP thanatin with New Delhi metallo-β- lactamase leads to hypersusceptibility to carbapenems [5]. Nisin and daptomycin are representative AMPs that damage the bacterial cytoplasmic membrane via complex formation with lipid II (a cell-wall precursor) [6] and with phosphatidylglycerol (a membrane phospholipid) as well as lipid II [7], respectively.

Multidrug-resistant bacteria are a serious global issue. In particular, drug-resistant *Enterococcus feacium*, *Staphylococcus aureus*, *Klebsiella pneumoniae*, *Acinetobacter baumanni*, *Pseudomonas aeruginosa*, and *Enterobacter* spp. (collectively known as ESKAPE) are problematic in nosocomial infections [8]. Some *S. aureus* strains are resistant to vancomycin and daptomycin, which are used against methicillin-resistant *S. aureus* (MRSA) [9,10]. Antibiotics targeting the biosynthetic pathways of biomolecules are generally effective in the exponential phase of bacterial growth. In contrast, AMPs that cause physical damage to the bacterial cytoplasmic membrane do so at all stages of cell growth, even in the stationary phase, so the probability of emergence of AMP-resistant bacteria is low. The mechanism of bacterial resistance to AMPs is due to the modification of cell surface structures, such as D-alanine modification of teichoic acid and lysine modification of phospholipids, degradation by extracellular proteases, or peptide efflux systems (e.g., ABC transporters) [11,12].

AMPs are crucial to the innate defense system of arthropods, particularly cecropins (from *Hyalophora cecropia*) [13] and insect defensins, which have been isolated from various species [14,15]. Tick defensins contribute significantly to maintaining the bacterial flora in the tick midgut and controlling pathogen infection [16,17]. It was reported that *Borrelia garinii* and *Stenotrophomonas maltophila* isolated from *Ixodes persulcatus* were resistant to tick defensins. Additionally, *S. aureus*, a Gram-positive pathogen of serious concern, was not identified in isolates from the tick midgut [16], suggesting that *S. aureus* growth may be restricted in the tick midgut due to antimicrobial factor(s).

Persulcatusin (IP), a mite AMP found in the midgut of *I. persulcatus*, exhibits antibacterial activity against Gram-positive bacteria, particularly *S. aureus* [18,19]. IP also exhibits antibacterial activity against multidrug-resistant *S. aureus*, including MRSA and vancomycin-resistant *S. aureus* (VRSA) [20,21]. Furthermore, scanning electron microscopy observation and calcein leakage assays suggest that the mechanism of action of IP is primarily membrane disruption and that IP may also have one or more unknown mechanisms [20]. However, the experimental investigations to date have been unable to determine how IP resistance develops and what factors are associated with its mechanism of action against *S. aureus*.

This study investigated whether IP resistance evolved with continued exposure to sublethal concentrations of IP. We also employed a transposon (Tn) insertion mutant library to screen clones with lower MICs of IP compared with the parent strain, and genes disrupted by Tn-insertion were identified as genes associated with IP resistance. Elucidation of the mechanism of IP resistance will provide insight into another unknown mechanisms of action of AMPs distinct from membrane disruption.

## 2. Materials and Methods

### 2.1. Antimicrobial Peptides and Antibiotics

The AMPs used in this study included IP (GFGCPFNQGACHRHCRSIGRRGGYCAGLFKQTCTCYSR [19]), bLfinB (FKCRRWQWRMKKLGAPSITCVRRAF [22]), which was derived from pepsin-digested bovine lactoferrin, the lantibiotic peptide nisin (MP Biomedicals, Santa Ana, CA, USA) and the lipopeptide daptomycin (Tokyo Chemical Industry, Tokyo, Japan). IP and bLfinB were chemically synthesized by Scrum Inc. (Tokyo, Japan) and GL Biochem Co., Ltd. (Shanghai, China), respectively. The antimicrobial agents used in this study included vancomycin hydrochloride (FUJIFILM Wako Pure Chemical Corp., Osaka, Japan) and cephazolin sodium salt (Tokyo Chemical Industry). IP was dissolved in dimethyl sulfoxide (DMSO) (FUJIFILM Wako Pure Chemical Corp.) to a concentration of 5 mg/mL and then diluted to a concentration of 160 µg/mL with sterile water. The formation of disulfide bridge was accomplished by air oxidation, as previously described [18]. Thereafter, the products were analyzed by electrospray ionization mass spectrometry using a micrOTOF-Q-II mass spectrometer (Bruker Daltonic, Bremen, Germany).

### 2.2. Bacterial Strains and Media

*S. aureus* strains ATCC 29213 and JE2 and the Tn-insertion mutant library derived from strain JE2 (NTML) [23] were used in this study. Mueller–Hinton broth (MHB) (Becton Dickinson, Sparks, MD, USA) and cation-adjusted MHB (caMHB) (Becton Dickinson) were used for MIC determination. Stock solutions of NTML strains were stored in 25% (*v*/*v*) glycerol at −70 °C until use and streaked onto a tryptic soya agar (TSA) plate (Nissui, Tokyo, Japan) containing 5 µg/mL erythromycin (FUJIFILM Wako Pure Chemical Corp.).

### 2.3. MIC Determination

The MIC of antimicrobial substance is defined as the lowest concentration with any visible bacterial growth. We determined the MICs using the broth microdilution method [24,25]. MHB supplemented with 0.1% bovine serum albumin (MHB-BSA), MHB, and caMHB was used to determine the MICs for CAMPs (IP, bLfinB, and nisin), vancomycin and cephazolin, and daptomycin, respectively. Briefly, a two-fold concentrate of each medium (50 μL) was placed into wells of a 96-well polypropylene round bottom plate (Watson Bio Lab, Tokyo, Japan). Next, serial two-fold dilutions of the appropriate antimicrobial agents prepared with sterile water (40 μL) were added to each well. Then, overnight bacterial cultures (37 °C with shaking at 120 rpm) were diluted to 5 × 10^6^ colony- forming units (cfu)/mL with sterile phosphate-buffered saline (PBS), and 10 μL was inoculated into each well. Finally, the plates were incubated at 37 °C for 22 h, after which the MICs were determined.

### 2.4. Time-Killing Assay

*S. aureus* strain ATCC 29213 cells cultured to the exponential growth phase (optical density at 660 nm [OD_660_], 0.5–0.6) in MHB were harvested following centrifugation at 5000× *g* for 5 min at 23 °C. Then, the bacterial cells were washed with sterile PBS and resuspended in two-fold concentrated MHB-BSA (for IP) or caMHB (for daptomycin) at an OD_660_ of 0.5. Next, the bacterial suspension and antimicrobial peptides (IP or daptomycin) were mixed in a ratio of 1:1 (*v*/*v*) at a final concentration of 10 × MIC of IP (10 μg/mL) or daptomycin (20 μg/mL). After incubation at 37 °C for 0, 15, 30, 60, 90, 120, and 150 min, the cells (0.5 mL) were collected following centrifugation at 5000× *g* for 5 min at 23 °C. The cells were then washed with 1 mL of sterile PBS, and appropriately diluted cells were spread onto TSA plates. After incubation at 37 °C overnight, the percentage of viable cells was calculated as CFU_sample_/CFU_time 0_ × 100.

### 2.5. Membrane Potential Assay

Membrane potential was measured by using a fluorescent assay based on the method of Silverman et al. [26]. Briefly, *S. aureus* ATCC 29213 cells prepared as described in the previous Section 2.4 were treated with 10 × MIC of IP (10 μg/mL) daptomycin (20 μg/mL), or nisin (2560 μg/mL). After incubation at 37 °C for 0, 15, 30, 60, 90, and 120 min, the cells (2 mL) were transferred to a fluorometer cuvette with a stirring rod at 400 rpm. Fluorescence was determined using an FP-8350 spectrofluorometer (Jasco Corp., Tokyo, Japan) with an excitation and emission wavelength of 622 nm and 670 nm, respectively. First, background data were collected for 30 s before adding a DiSC_3_(5) (3,3’-dipropylthiadicarbocyanine iodide) (Anaspec, Fremont, CA, USA). This fluorescence probe prepared in DMSO was added to a final concentration of 1 μM into the cuvette with stirring at 400 rpm. Then, the fluorescence intensity data were collected for an additional 4.5 min. The membrane potential as a percentage of the control was calculated as follows: [(max/min)_sample_/(max/min)_time 0_] × 100, where max was the maximum fluorescent signal in the trace (<10 s after dye addition) and min was the minimum signal (the average signal over the last 10 s of the trace). 

### 2.6. Membrane Permeability Assay

Membrane permeability was measured using a fluorescence assay based on the method of Silverman et al. [26]. Briefly, *S. aureus* ATCC 29213 cells prepared as described in Section 2.4 were treated with 10 × MIC of IP (10 μg/mL) daptomycin (20 μg/mL), or nisin (2560 μg/mL). After incubation at 37 °C for 0, 15, 30, 60, 90, and 120 min, the fluorescent probe propidium iodide (PI) (FUJIFILM Wako Pure Chemical Corp.), which had been previously dissolved in sterile water, was added to a final concentration of 10 μM. Then, the mixture (2 mL) was transferred to a fluorometer cuvette with a stirring rod at 400 rpm. The fluorescence intensity was determined using an FP-8350 spectrofluorometer at excitation and emission wavelengths of 535 nm and 617 nm, respectively, monitoring the fluorescence for 5 min with stirring at 400 rpm. The fluorescence intensity at 617 nm was defined as the average signal of the last 10 s of the curve.

### 2.7. Serial Passaging of S. aureus Cultured with IP

Serial passage cultures of *S. aures* ATCC 29213 and individual antimicrobials (IP, nisin, daptomycin, vancomycin, and cefazoline) were performed. The inoculum was *S. aureus* ATCC 29213 cultured in MHB diluted to an OD_660_ of 0.025 (approximately 1 × 10^7^ cfu/mL) with sterile PBS. Serial passage culture was initiated by adding 10 μL of inoculum to 1 mL of MHB-BSA, ca-MHB, or MHB containing AMPs (IP and nisin), daptomycin, or vancomycin and cephazolin, respectively, at 1/2 × MIC and 1 × MIC for *S. aureus* ATCC 29213. The cells were then cultured by shaking (120 rpm) at 37 °C overnight. The next day, the cultured cells (OD_660_ ≥ 0.5) were diluted to an OD_660_ of 0.025 with sterile PBS, and 10 μL of the dilution was used for the next passage. If bacterial growth was only observed at 1/2 × MIC, the next passage was performed at concentrations of 1/2 × MIC and 1 × MIC. If bacterial growth was observed in the presence of 1 × MIC of the respective antimicrobial agent, the next passage was performed at a concentration of 1 × MIC and 2 × MIC. This procedure was repeated for 20 cycles, and the bacterial cells at each passage step were collected and stored in 25% (*v*/*v*) glycerol at −70 °C. Three independent clones from passage 5, 10, 15, and 20 were recovered, and the MICs of the respective antimicrobial agents for these isolates were determined using the broth microdilution method described in Section 2.3.

### 2.8. Fitness Measurements

Overnight cultures at 37 °C of *S. aureus* in MHB were inoculated into 5 mL of fresh MHB (OD_600_, 0.001). Growth was monitored at 37 °C with shaking at 150 rpm for 20 h in a Bioshaker BR-43FL (Taitec, Tokyo, Japan) equipped with an ODBox-C (Taitec) by measuring the OD_600_ every 30 min. The relative growth rate was calculated based on OD_600_ values during the exponential growth phase.

### 2.9. Screening of S. aureus Variants Showing IP Hypersusceptibility

We used the NTML, containing 1920 Tn-insertion mutants, to identify any factors associated with the antimicrobial action of IP. Each mutant underwent standing culture for 20 h at 37 °C in the well of a 96-well plate containing 100 μL MHB. Then, the precultured cells were diluted 100-fold in MHB at a ratio of 1:9, and 10 μL of each dilution was inoculated into 90 μL MHB-BSA containing a final concentration of 0.5 µg/mL IP (equivalent to 1/2 × MIC of their parent strain *S. aureus* JE2). After incubation at 37 °C for 20 h, we selected the Tn-insertion mutant strains that were unable to grow under the culture conditions. We then determined the MIC of IP, bLfinB, nisin, daptomycin, and vancomycin against these selected mutant strains using the broth microdilution method described in Section 2.3.

## 3. Results

### 3.1. Prolonged Exposure of IP Induces the Emergence of IP-Resistant Phenotypes with Increased MIC

IP is a member of the tick defensin family and contains six cysteine residues in its amino acid sequence [19]. When chemically synthesized IP was oxidized by air oxidation for 7 days, its molecular mass of 6 was reduced compared with that of the linear (unoxidized) form (Appendix A). This suggested that the oxidized IP had three disulfide bridges. The antimicrobial activity of oxidized IP was increased compared with that of the original linear IP (Appendix A), which was in good agreement with the results of a study published in 2011 [18]. Studies on known AMPs, including magainin 2 and gramicidin D or synthetic AMPs, have reported that bacteria are less likely to acquire resistance to AMPs that damage the bacterial cytoplasmic membrane compared with antibiotics such as rifampicin and ciprofloxacin [27,28]. To investigate this further, we performed a serial passage experiment in the presence of sub-minimum inhibitory concentration (MIC) levels of IP to generate *S. aureus* spontaneous mutants with increased MIC IP. Additionally, to verify the occurrence of cross-resistance between IP and other antimicrobial agents, strains with increased MICs of other antimicrobial agents (nisin, daptomycin, and vancomycin) whose mechanism of action may be related to that of IP were also generated.

In *S. aureus* spontaneous mutants from the 20th serial passage, the MIC of IP was 4-fold higher (4 µg/mL) than that before serial passaging (1 µg/mL) (Figure 1A). This increased MIC of IP was similar to that observed with vancomycin (4-fold, 4 µg/mL) (Figure 1B). Furthermore, the AMPs nisin and daptomycin showed increased MICs of ≥32-fold (≥4096 µg/mL) and 16-fold (32 µg/mL), respectively, compared with the MICs in untreated cells (Figure 1C,D). These findings suggested that *S. aureus* was less likely to acquire resistance to IP than to other AMPs. The antimicrobial agent that was the most vulnerable to resistance among those tested was cephazolin, a commonly used β-lactam antibiotic, where the MIC increased ≥200-fold (>64 μg/mL) (Figure 1E). The negative controls (absence of antimicrobial agents) showed no change in the MICs of the corresponding antimicrobial agents under the test conditions (Figure 1A–E).

### 3.2. S. aureus Mutant with Increased MIC of IP Reduces Susceptibility to Other AMPs and Vancomycin

Generally, cross-resistance is observed between two antimicrobial agents if they have a similar mechanism of action and/or resistance. Thus, we evaluated the susceptibility of IP-resistant mutants to other antimicrobial agents. The IP-resistant mutant strain (IPR) isolated from the 20th serial passage with IP showed 2-fold higher MICs of nisin, daptomycin, and vancomycin than wild-type (WT) ATCC 29213 (Table 1). On the other hand, no change was observed in the MICs of the other antibiotics tested (i.e., cephazolin, gentamycin, and ciprofloxacin) (Table 1). Additionally, the nisin-resistant mutant strain (NIR) and daptomycin-resistant mutant strain (DPR), isolated from the 20th serial passage with nisin and daptomycin, respectively, showed 4-fold higher MICs of IP, which were equivalent to those of IPR (4 µg/mL) (Table 1). Interestingly, the vancomycin-resistant mutant strain (VCR) and cephazolin-resistant mutant strain (CER), isolated from the 20th serial passage with vancomycin and cephazolin, respectively, displayed lower MICs of IP (0.5 and <0.5 µg/mL, respectively) (Table 1). Notably, VCR showed a 4-fold higher MIC of nisin, suggesting that the action of IP differed from that of nisin and vancomycin.

### 3.3. IP Resistance Leads to Small Colony Variant Phenotype

During the course of the serial passage experiment, we observed that colonies of IPR and NIR cultured on sheep blood agar plates were smaller compared with colonies of the WT, VCR, and DPR strains, and their characteristics were similar to small colony variants (Figure 2A). Small colony variants were characterized by a significant reduced growth rate and increased doubling time [29]. Thus, we monitored the growth and calculated doubling time in liquid medium of the IPR strains and other antimicrobial agent resistant mutants. The doubling times of the IPR and NIR strains were significantly longer than that of the WT, while the doubling times of the VCR and DPR strains fell between those of the WT and IPR strains (Figure 2B).

### 3.4. IP Kills S. aureus by Depolarization and Slow Permeabilization of the Bacterial Cytoplasmic Membrane

Generally, AMPs, including daptomycin, damage bacterial cytoplasmic membranes. Therefore, we investigated the mechanism of action of IP on the *S. aureus* strain ATCC 29213 using a fluorescent probe (Figure 3A,B). IP rapidly decreased the cytoplasmic membrane potential by approximately 50% after 15 min, and daptomycin decreased the cytoplasmic membrane potential by <50% after 60 min (Figure 3A). The membrane potential decreased to approximately 20% after 120 min for both IP and daptomycin (Figure 3A). The fluorescence intensity at 617 nm, an indicator of membrane permeability, increased over time for both IP and daptomycin (Figure 3B). However, cells treated with IP showed a smaller loss in membrane permeability than cells treated with daptomycin (Figure 3B). Correspondingly, IP decreased the cell viability by 28% after 60 min and daptomycin decreased cell viability by 80% after 15 min (Figure 3C).

### 3.5. Selection of Tn-Insertion Mutant Strains Showing Hypersusceptiblity to IP

The above-mentioned findings suggested that the mechanism of resistance in IP was primarily associated with bacterial cytoplasmic membrane damage and that IP also had an unknown mode of action that probably differed from that of nisin and daptomycin. Conditional lethal phenotypes (in bacteria) are an extremely useful phenomenon for studying genes essential for growth, searching for new targets for novel antibiotic discovery, and providing experimental demonstration of the mode of antibiotics [30,31]. Therefore, we used the Nebraska Transposon Mutant Library (NTML) to select mutants showing hypersusceptibility to IP in the presence of sublethal IP concentrations. Consequently, 16 mutant strains showing hypersusceptibility toward IP were identified, and their MICs were 2–16-fold lower than that of their parent strain (*S. aureus* JE2). The genes identified were associated with resistance to the cationic AMPs (CAMPs) nisin and daptomycin and to other cell-wall damaging antimicrobial agents, such as vancomycin and oxacillin. We classified them into three groups according to the disrupted genes (Table 2).

Group 1 included Tn-insertion mutants showing hypersusceptibility to CAMPs and daptomycin. These six mutants, namely strains NE1360, NE675, NE70, NE481, NE1756, and NE592, possessed the inactivated genes *fmtC* (equivalent to *mprF*), *vraF*, *vraG*, *graR*, and *graS*, and *atpA*, respectively, which have been reported to be associated with CAMP resistance [32,33,34]. Among them, strains NE1360, NE675, NE481, and NE1756 showed markedly increased IP susceptibility with 8–16-fold lower MICs of IP. Additionally, these strains also showed hypersusceptibility to other CAMPs such as bovine lactoferricin B (bLfinB), lipopeptide (daptomycin), and lantibiotic peptide (nisin) (Table 2). Strains NE70 and NE592 showed slightly higher susceptibility to IP (2–4-fold lower MICs), daptomycin (2-fold lower MICs), and nisin (2–4-fold lower MICs) (Table 2). 

Group 2 included Tn-insertion mutants showing hypersusceptibility to bacitracin and nisin. These six mutants, namely strains NE1766, NE1105, NE1768, NE775, NE1116, and NE890, possessed the inactivation genes *braD*, *braE*, *vraD*, *vraE*, *braS*, and the gene adjacent to *braR* (SAUSA300_2560), respectively, and these genes are associated with bacitracin and nisin resistance [35]. The mutants in Group 2 showed hypersusceptibility not only to nisin but also to IP. Notably, strains NE1766, NE1105, and NE1116 showed markedly higher susceptibility to IP (8–16-fold lower MICs) and nisin (4-fold lower MICs) compared with that of their parent strain (Table 2).

Group 3 included Tn-insertion mutants associated with resistance to antimicrobial agents that inhibit bacterial cell-wall biosynthesis. The four strains in this group all showed hypersusceptibility to IP and possessed a Tn-insertion in the genes associated with resistance to cell-wall active antimicrobial agents such as vancomycin and β-lactams. Strains NE554 and NE823 possessed a Tn-insertion in *vraR* and *vraS*, respectively, which encode VraSR, a two-component system (TCS) involved in vancomycin resistance [36]. These strains showed slightly higher susceptibility to IP (4-fold lower MICs) and vancomycin (2-fold lower MICs) (Table 2). Interaction between the GraRS TCS and the VraFG ABC transporter was reported to be involved in vancomycin resistance in *S. aureus* [37], and the mutant strains NE675, NE481, and NE1756 possessed inactivated *vraF*, *graS*, and *graR* genes, respectively, and showed slightly higher susceptibility to vancomycin. Additionally, strains NE1022 and NE0980 possessed inactivated *fmt* (equivalent to *fmtA*) and *auxA* genes, respectively. These genes were reported to be associated with oxacillin resistance [38,39] and showed 8-fold and 2-fold lower MICs, respectively, of IP compared with their parent strain (JE2) (Table 2).

## 4. Discussion

Antibiotics inhibit the synthesis of essential bacterial biocomponents, whereas AMPs act on the bacterial cytoplasmic membrane. Therefore, resistance to AMPs is assumed to be less likely to occur due to genetic mutation. This is supported by the fact that resistance to the AMPs is less likely to occur than resistance to conventional antibiotics, such as quinolones and rifampicin [27,28]. In our study, the rate of emergence of mutants that show IP-resistant phenotypes (increased MIC) was significantly less than the rate of emergence of mutants resistant to cephazolin, which inhibits bacterial cell-wall synthesis, although it was comparable to vancomycin (Figure 1). Additionally, mutants with IP-resistant phenotypes (with increased MIC of IP) were less likely to emerge compared with mutants resistant to nisin and daptomycin (Figure 1). Furthermore, although occurring infrequently, prolonged exposure to sublethal levels of AMPs resulted in mutant strains that were resistant to IP (showed increased MIC of IP) as well as other AMPs (Figure 1), and IPR showed significantly reduced growth rates compared with DPR and VCR (Figure 2). Mutant strains of *S. aureus* showing resistance to AMPs such as human AMP LL-37 and insect AMP tenecin-1 have also been reported to exhibit a reduced growth rate [40,41]. These data suggest that resistance to IP (increase in the MIC of IP) in *S. aureus* is a significant burden on growth due to gene mutation(s) and gene expression, as observed in other AMPs [40,41].

Analysis of cross-resistance showed that mutants resistant to each AMP, including IPR, were simultaneously resistant to other AMPs, such as nisin and daptomycin, and to vancomycin. However, IPR showed a different pattern of cross-resistance to other antimicrobial agents, such as vancomycin and cephazolin, compared with the other AMP-resistant mutants (Table 1), suggesting that the mechanism of action of IP differed from other AMPs, such as nisin and daptomycin. 

Our investigation of the mechanism of antimicrobial activity of IP against *S. aureus* using a fluorescent probe (Figure 3A,B) revealed that IP targets the bacterial cytoplasmic membrane, leading to depolarization, as was the case with daptomycin. However, membrane permeabilization caused by IP was significantly slower compared with membrane permeabilization caused by daptomycin (Figure 3A,B). Thus, our finding suggested that the mechanism of action of IP differed from that of daptomycin. Notably, IP is effective against MRSA and VRSA [20], whose mechanisms of resistance are PBP2*′* acquisition [42] and alteration of the cell-wall component D-alanyl-D-alanine to D-alanyl-D-lactic acid [43], respectively. This also supports the possibility that IP may have a different target for its mechanism of action compared with that of β-lactam and vancomycin.

We screened a gene-deficient mutant library to identify factors associated with IP resistance (increased MIC of IP) and identified 16 mutants with hypersusceptibility to IP (Table 2). These were classified into three groups based on previous reports (Figure 4). Although the 16 genes identified in this study have been previously reported, they are presumed to play a role in IP resistance (the increased MIC of IP) through a mechanism that differs from other AMPs. FmtC (MprF), the GraSR TCS, and VraFG have been identified as resistance factors in other AMPs, such as LL-37 and human defensins [11,12,33]. FmtC (MprF), which modulates bacterial cytoplasmic membrane charges due to the formation of lysylphosphatidylglycerol (LPG), confers resistance on bacterial cells against β-lactams and CAMPs [44,45,46]. The GraSR TCS controls CAMP resistance in *S. aureus* through D-alanylation of bacterial cell-wall teichoic acid, which is mediated by the *dltABCD* operon, and the LPG formation by FmtC [47,48]. This TCS also controls the expression of the vraFG operon, located directly downstream of GraSR genes, which encodes an ABC transporter contributing to CAMP resistance [33]. These genes were also selected as factors associated with IP resistance (the increased MIC of IP), indicating that the mechanism of action and/or resistance to IP (the increased MIC of IP) is likely shared, at least in part, with these AMPs. It is noteworthy that FmtC (MprF) and VraFG are found to be present in the core genome of *S. aureus* [49], suggesting that IP has a potential to be a novel anti-*S. aureus* therapeutic agent.

In terms of functions related to cell-wall biogenesis, the TCS BraRS was reported to be involved in bacitracin and nisin resistance by modulating ABC transporters BraDE and VraDE [35]. Additionally, VraDE was involved in resistance to daptomycin vancoymcin, and human AMPs, such as HBD-3 and LL-37, as well as bacitracin and nisin [50]. Another TCS, VraRS, was identified as a factor involved in vancomycin and β-lactam resistance via modulation of cell-wall biosynthesis genes [36]. Fmt (FmtA), the mechicillin resistance factor, was reported to modulate the physical property (charge) of teichoic acid by hydrolation of ester bonds between D- alanine and the backborne of tehicoic acids [38,51]. Furthermore, *fmtA* inactivation rendered *S. aureus* cells hypersusceptible to daptomycin and vancomycin [51]. AuxA was reported to modulate β-lactam susceptibility in MRSA by stabilizing lipoteichoic acid [39]. Since all of these factors are associated with cell-wall biogenesis and stabilization, it is reasonable to speculate that the cell-wall metabolic pathway may be a target(s) of IP.

The patterns of IP cross-resistance (Table 2) and MICs for the 16 Tn-insertion mutants (Table 2) were similar to those of nisin. However, it is interesting to note that VCR showed an opposite susceptiblity to IP and nisin of 0.5 μg/mL and a 4-fold higher MIC, respectively, compared with their parent strain (Table 1). Additionally, nisin exhibited a strikingly strong impact on the membrane potential and membrane permeability compared with IP (Appendix A), suggesting that the modes of action of IP and nisin are, in part, common, but IP is functionally different compared with nisin.

Our study results led us to speculate about the mechanism of action of IP. First, IP is adsorbed on the bacterial surface by electrostatic interaction with negatively charged membrane phospholipids, such as phosphatidylglycerol and cardiolipin, and this interaction depolarizes the bacterial cytoplasmic membrane. The resultant change in membrane potential alters the membrane structure by hydrophobic interaction with the membrane lipid tail, which is probably insufficient to cause membrane damage, although this interaction may delocalize membrane proteins associated with cell-wall biosynthesis and stabilization, thereby inhibiting the cell-wall metabolic pathway. Further detailed analysis is necessary to elucidate the mechanism of action of IP. 

## Figures and Tables

**Figure 1 microorganisms-12-00412-f001:**
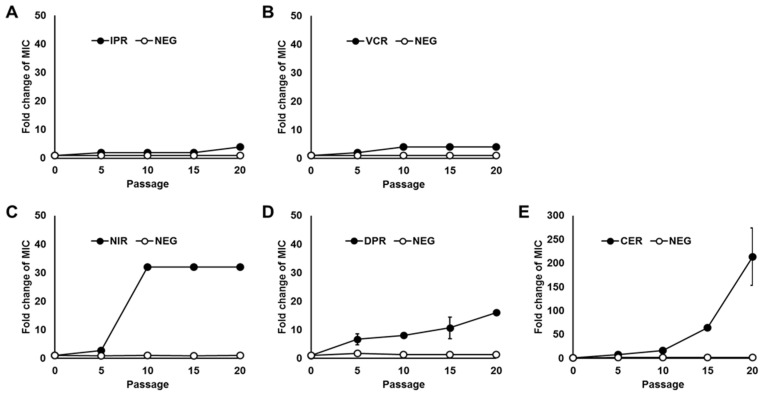
Incidence of bacterial resistance to antimicrobial agents. Fold change in MIC values of isolates from each serial passaging culture with increasing concentrations of antimicrobial agents: (**A**) IP-resistant, IPR; (**B**) vancomycin-resistant, VCR; (**C**) nisin-resistant, NIR; (**D**) daptomycin-resistant, DPR; and (**E**) cephazolin-resistant, CER) (black), and without these antibiotics (NEG, white) compared with parent strain *S. aureus* ATCC 29213. MICs of IP (1 µg/mL), VCM (1 µg/mL), NIS (256 µg/mL), DAP (2 µg/mL), and CEZ (1 µg/mL) for *S. aureus* ATCC 29213 are shown in Table 1. Results were the averages of three independent clones and error bars indicated SD.

**Figure 2 microorganisms-12-00412-f002:**
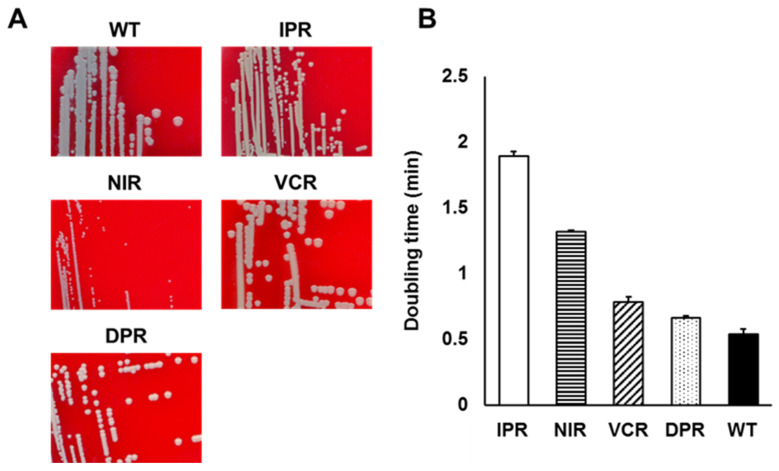
(**A**) Colony phenotype of each strain. Cells were grown on the sheep blood agar plates and incubated at 37 °C for 24 h. (**B**) The doubling time of each antimicrobial-agent-resistant mutants. Cells such as the IPR, NIR, VCR, DPR strains, and WT were grown in MHB medium, and the doubling times were calculated based on the optical density at 600 nm during the exponential growth phase.

**Figure 3 microorganisms-12-00412-f003:**
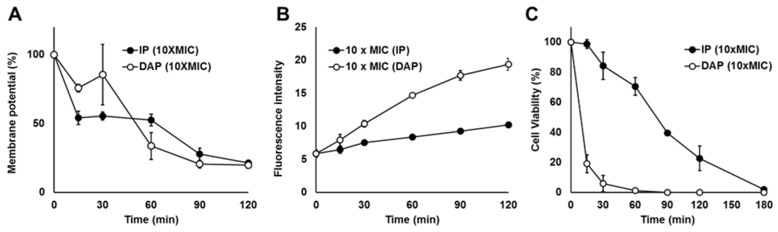
Effect of IP on the cell membranes and cell viability. For the analysis of (**A**) bacterial cytoplasmic membrane potential, (**B**) bacterial cytoplasmic membrane permeability and (**C**) cell viability of *S. aureus* ATCC 29213, cells were grown in Mueller–Hinton broth (MHB) medium and treated with IP (black) or daptomycin (DAP, white). Membrane potential (A) was determined as a percentage of 0 time and calculated by [(max/min)_sample_/(max/min)_0 time_] × 100%, where max was defined as the maximum fluorescent intensity at 670 nm (excited at 622 nm) in the trace (<10 s after dye addition) and min was defined as the minimal intensity (the average intensity of the last 10 s of the trace). Membrane permeability (**B**) was defined as the average intensity at 617 nm (excited at 535 nm) of the last 10 s of the trace.

**Figure 4 microorganisms-12-00412-f004:**
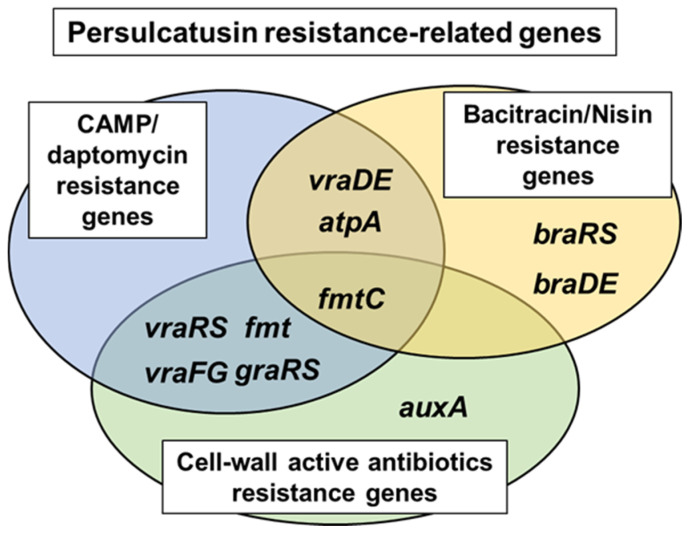
An overview of the relationship between the identified genes related to IP resistance and other AMP resistance genes. Genes in the blue circle are involved in resistance to CAMPs and daptomycin. Genes in the yellow circle are involved in resistance to bacitracin and nisin. Genes in the green circle are involved in resistance to cell-wall active antimicrobial agents, such as oxacillin and vancomycin.

**Table 1 microorganisms-12-00412-t001:** Cross-resistant analysis of antimicrobial resistant mutants.

		* MIC (μg/mL)						
Strain	Relevant Description	IP	NIS	DAP	VAN	CEZ	GEN	CIP
ATCC 29213	Wild type	1	256	2	1	1	0.5	0.25
IPR	Persulcatusin-resistant	4	512	4	2	1	0.5	0.25
NIR	Nisin-resistant	4	>4096	4	2	0.5	0.25	0.25
DPR	Daptomycin-resistant	4	256	32	2	0.5	0.5	0.25
VCR	Vancomycin-resistant	0.5	1024	4	4	1	0.25	0.25
CER	Cephazolin-resistant	<0.5	64	2	0.5	>64	<0.125	0.25

* IP, persulcatusin; NIS, nisin; VAN, vancomycin; DAP, daptomycin; CEZ, cephazolin; GEN, gentamycin; CIP, ciprofloxacin.

**Table 2 microorganisms-12-00412-t002:** MICs of antimicrobial agents against IP-susceptible Tn-insertion mutants.

		* MIC (μg/L)				
		CAMPs		Lipopeptide	Lantibiotic	Glycopeptide
Strain	Relevant Description	IP	LfinB	DAP	NIS	VAN
JE2	Wild type	1	>64	4	128	1
Group 1. CAMP/daptomycin-resistant						
NE1360	∆*fmtC* (SAUSA300_1255)	0.125	16	1	32	1
NE675	∆*vraF* (SAUSA300_0647)	0.25	8	1	32	0.5
NE70	∆*vraG* (SAUSA300_0648)	0.5	>64	2	64	1
NE481	∆*graR* (SAUSA300_0645)	0.25	4	1	32	0.5
NE1756	∆*graS* (SAUSA300_0646)	0.25	4	1	32	0.5
NE592	∆*atpA* (SAUSA300_2060)	0.25	>64	2	32	0.5
Group 2. Bacitracin/nicin-resistant						
NE1766	∆*braD* (AUSA300_2557)	0.25	>64	2	32	1
NE1105	∆*braE* (SAUSA300_2556)	0.25	>64	4	32	1
NE1768	∆*vraD* (SAUSA300_2633)	0.5	>64	2	32	1
NE775	∆*vraE* (SAUSA300_2634)	0.5	>64	4	32	1
NE1116	∆*braS* (SAUSA300_2558)	0.125	>64	4	32	1
NE890	(SAUSA300_2560)	0.5	>64	2	64	1
Group 3. Cell-wall active antimicrobial-agent-resistant						
NE554	∆*vraR* (SAUSA300_1865)	0.25	>64	4	64	0.5
NE823	∆*vraS* (SAUSA300_1866)	0.5	>64	4	64	0.5
NE1022	∆*fmt* (SAUSA300_0959)	0.25	>64	2	64	0.5
NE0980	∆*auxA* (SAUSA300_0980)	0.5	>64	2	64	1

* CAMPs, cationic antimicrobial peptides; IP, persulcatusin; LfinB, bovine lactoferricin B. DAP, daptomycin; NIS, nisin; VAN, vancomycin.

## Data Availability

Raw data were generated at Tohoku University. Data supporting the findings of this study are available from the corresponding author H.Y. upon request.

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
