# Peer review of "Identification of Genes Associated with Resistance to Persulcatusin, a Tick Defensin from *Ixodes persulcatus"

_microorganisms, 2024, doi:10.3390/microorganisms12020412_

Round 1
Reviewer 1 Report
Comments and Suggestions for Authors
In the manuscript titled 'Identification of genes associated with resistance to persulcatusin, a tick defensin from Ixodes persulcatus' the authors generated spontaneously resistant mutants to persulcatusin and analyzed their cross-resistance to other antimicrobial peptides and antibiotics. They also used fluorescent probes to investigate the target of persulcatusin activity by evaluating persulcatusin-induced damage to the bacterial cytoplasmic membrane. However, even manuscript is technically well written and planned, there is nothing new and with the significance of content to the scientific community, to be able to suggest the acceptance of the manuscript in such a reputable journal.
Author Response
Thank you very much for taking your time to review our manuscript. Although the reviewer stated that this study is “nothing new”, persulcatusin is a rather new antimicrobial peptide obtained from a tick Ixodes persulcatus. Thus, its in-depth experiments are scarce as compared to other well-known antimicrobial peptides. More importantly, IP exhibits strong antibacterial activity against multidrug-resistant S. aureus, including MRSA and vancomycin-resistant S. aureus (VRSA), which are problematic pathogens. Antimicrobial resistance is a serious and global public health problem. To address this problem, it is important to find and develop now antimicrobial agents, which has a low probability to develop resistant mutants. We believe, in this sense, that this study provide valuable results and information to the scientific community for future development of novel antimicrobial agents.
Reviewer 2 Report
Comments and Suggestions for Authors
The work is interesting and deserves publication. The authors of the paper (Identification of genes associated with resistance to persulcatusin, a tick defensin from Ixodes persulcatus) demonstrated that IP is adsorbed on the bacterial surface by electrostatic interaction and this interaction depolarizes the bacterial membrane. The authors suggest that the antimicrobial activity of IP on bacterial cytoplasmic membranes occurs via a mechanism of action different from that of known AMPs. The authors identified 16 genes involved in IP resistance. I think that for the authors will be interested the paper to consider their results “Comparative Analysis of Proteomes of a Number of Nosocomial Pathogens by KEGG Modules and KEGG Pathways” in DOI: 10.3390/ijms21217839.
Comments on the Quality of English Language
I have no comments
Author Response
Thank you very much for taking your time to review our manuscript and positive comment and suggestion that helped improve our manuscript. We added a description by referring the paper as suggested. Please see lines 387-391.
Reviewer 3 Report
Comments and Suggestions for Authors
The manuscript entitled "Identification of genes associated with resistance to persulcatusin, a tick defensin from Ixodes persulcatus" described in depth the mechanism of activity and resistance of the antimicrobial peptide called IP.
There is only one comment / suugestion for the authors
it is not appriated to talk about IP resistance,due that susceptibility or resistance is a clinical interpretation.
As IP is not used in clinical and the authors do not Know thw pharmacoolgy of the compund ( efficacy, pharmacodynamics, pharmacokinetics et...) the IP antimicrobial peptide did not have anydefined berackpoint , it is better to talk about an increase of the MIC
Author Response
Thank you very much for taking your time to review our manuscript and positive comment and suggestion that helped improve our manuscript. We carefully examined and revised the original manuscript according to your comment/suggestion to avoid confusion by using phrases such as “increased MIC of IP”. Please see lines 19-20, 74-76, 190-191, 201-209, 227-228, 231-238, 342-343, 345-346, 359, 343, 373, 377, and line 387-391
Round 2
Reviewer 1 Report
Comments and Suggestions for Authors
After all changes proposed by another two reviewers, the manuscript can be published.